# Pathways of TB Transmission in Children—A Systematic Review of Molecular Epidemiological Studies

**DOI:** 10.3390/ijerph20031737

**Published:** 2023-01-18

**Authors:** Roland Diel, Albert Nienhaus

**Affiliations:** 1Institute for Epidemiology, University Medical Hospital Schleswig-Holstein, 24105 Kiel, Germany; 2LungClinic Grosshansdorf, Airway Research Center North (ARCN), German Center for Lung Research (DZL), 22927 Großhansdorf, Germany; 3Competence Center for Epidemiology and Health Services Research for Healthcare Professionals (CVcare), Institute for Health Services Research in Dermatology and Nursing (IVDP), University Medical Center Hamburg-Eppendorf (UKE), 20246 Hamburg, Germany; 4Department for Occupational Medicine, Hazardous Substances and Health Sciences (AGG), Statutory Accident Insurance and Prevention in the Health and Welfare Services (BGW), 22089 Hamburg, Germany

**Keywords:** tuberculosis, molecular epidemiology, genotyping, cluster, transmission, children

## Abstract

The widespread paradigm that younger children usually do not transmit *M. tuberculosis* complex (Mtbc) to their contacts has not yet been proven by genotypically confirmed transmissions. Therefore, we undertook a systematic review of molecular-epidemiological studies to investigate documented source and secondary TB (tuberculosis) cases among children. We searched the literature published before August 2022 using *PubMed*, *Cochrane*, and *Google Scholar* databases. PRISMA statement was used for systematic review. Of 312 records retrieved, 39 studies including children aged below 15 years offered epidemiological links between cluster members. In the 39 studies from 16 countries, 225 children were reported as cluster members of whom the overwhelming majority were infected by adults. Only 3 children—of those were 2 children aged below 10—were reported to be the definite source cases of 11 other children and 1 adult with genotypically matched Mtbc isolates. To date, molecular-epidemiological studies involving children with verified transmission links are scarce. As far as the heterogeneity of the studies we identified allows, we could conclude that the results confirm the paradigm that children aged below 10 hardly ever transmit Mtbc to others. The true extent of TB transmission through children may, however, be underestimated by those selected studies.

## 1. Introduction

Children, especially those aged below 5 years, who are contacts of sputum-smear-positive pulmonary TB cases are at high risk of developing active TB [1]. In 2020, about 1.1 million children, i.e., 11% of the world’s estimated total of 10 million cases, fell ill with TB [2]. Beyond this, the question remains as to whether children, once diagnosed with active TB, must then be considered as potential sources of further TB cases, i.e., whether they may contribute to the spread of TB disease.

DNA genotyping denotes molecular typing of an Mtbc strain and is helpful in determining whether “fresh” transfer by an index person to another contact person has occurred. This can be documented by the identification, within months or a few years, of an index patient’s specific Mtbc strain in the sputum of a person with whom that index person, while ill with TB, had close contact. The forming of a “cluster”, i.e., the identification of a particular Mtbc strain affecting at least two different patients, is always the result of a two-stage process: in the first stage, an infection occurs by transfer from the diseased index person to the follow-up person as primary infection or secondary reinfection, at first causing no illness. In the second step, this latent infection must progress to manifest TB disease. Principally, Mtbc transmission can be measured in two ways: centripetally (from the diseased person back to the presumed source, i.e., a previous infectious pulmonary disease case with which the now-diseased person had contact) and centrifugally (from the now-diseased index case forward towards contacts who may have been latently Mtbc-infected by that person or who even, in the meantime, have developed TB disease from that infection).

Under the current German guidelines [3], and following the conventional international paradigm, children below 10 years of age are as a rule not generally considered possible source cases requiring inclusion in centripetal contact tracing. The reason for this scientific doctrine is that patients in this age group are considered rarely contagious given the general characteristics of their disease: low bacilli load, weak coughing, and a smaller number of cavities [4,5,6,7,8].

Despite this, numerous case reports have been published in the past decades reporting Mtbc transmission by children to other children or even to their adult contacts (e.g., [9,10,11,12]). Most of these findings are based on the results of tuberculin testing or TB blood tests (interferon-gamma release assays. IGRA) in contacts according to the “stone-in-the-pond” scheme, but only in very few reports are genotyped results of matched Mtbc isolates available. As it may be difficult to distinguish between fresh and remote latent TB infections, especially when the contacts of a child suffering from TB are only casual, cluster investigations looking for genetically identical Mtbc isolates are decisive in certifying that a contact person, having progressed in the meantime to culture-confirmed TB, was indeed infected by the suggested source case.

Thus, with the goal of providing more comprehensive evidence on the role of children in Mtbc transmission verified by cluster analysis and to confirm or disprove the prevailing view that infectiousness of children is low, we conducted an up-to-date systematic review that includes the most recently published studies without restriction to any intervention type, taking a qualitative approach. The review focuses on both TB source cases and secondary cases.

## 2. Materials and Methods

### 2.1. Study Search and Selection Criteria

We searched the literature published before August 2022 using *PubMed*, *Cochrane*, and *Google Scholar* databases. PRISMA statement was used for systematic review. We considered all studies reporting genotypical clusters with epidemiologically verified Mtbc transmissions among children aged below 15 years, according to the definitions below, irrespective of the genotyping method used. Children had to be clearly distinguished from adolescents or adults aged above 15 years and reported to be cluster members. Beyond presentation of sociodemographic and clinical data, epidemiological links between the cluster members had to be shown explicitly or could at least be retrieved from the text.

Keywords in these database searches included children, tuberculosis, *Mycobacterium tuberculosis*, transmission, spread, genotyping and cluster (full search string is detailed in Appendix B). Studies had to present original data on *M. tuberculosis*; reviews, conference contributions, editorials, letters, modelling articles, guidelines, and recommendations were excluded a priori.

### 2.2. Definitions Used for Study Analysis

Clusters were defined as at least two patients whose strains were found to be of the same *M. tuberculosis* genotype according to the genotyping procedure chosen in each study. Epidemiological links were defined as patients in a cluster identifying another patient in the same cluster as a definite contact confirmed by public health investigations. A source case was defined as the TB case with culture-confirmed pulmonary TB that had led to a secondary case with the same genotype and a traceable epidemiological connection. Previously used simplifications, such as considering a cluster member whose date of diagnosis appears first in a cluster as a source case of the subsequent patient (n-1 method), were not applied.

### 2.3. Data Extraction

Two reviewers (R.D. and A.N.) independently screened all titles identified in the database searches following a three-stage procedure: first were titles alone, then abstracts of the selected titles, and finally a full-text review of the selected abstracts. Any discrepancies were resolved by consensus.

### 2.4. Variables Assessed

The following variables were recorded, if available: (1) country of study and year of publication; (2) study period; (3) type and characterization of study; (4) type of genotyping; (5) number of participants initially included; (6) number of culture-confirmed study participants whose isolates were finally genotyped; (7) number of children included; (8) number of children with pulmonary TB (as far as reported); (9) number of reported clusters; (10) number of cluster members; (11) number of clusters in which children were included; (12) number of children in clusters; (13) assessment of whether contact tracing had been performed in addition to the cluster analysis (yes/no); (14) number of transmissions from source cases to other cluster members; (15) source cases of transmissions; (16) assessment of whether a child could be considered; and (17) assessment of drug resistance (if any).

## 3. Results

Figure 1 presents a flow diagram of the literature search results. In the selection process, 314 journal abstracts in English were identified. After 31 records were excluded based on their abstracts, a total 283 abstracts were read in full text.

In 25 of the studies, no children were included, and in 13 that included children, no epidemiological data were available. Six studies did not reveal the data on the children’s age, which would have been required to differentiate children from adults and older children from the subgroup of children aged below 10 years. In nine studies, no genotyping was performed, and in eight studies, despite genotyping, no clusters were reported. In 44 studies, sociodemographic or clinical data on children were available before genotyping was conducted, but no children were reported as cluster members. In 94 studies that reported children as cluster members, transmission links in clusters were not searched, or if there were any, no information was stated. Genetic studies (n = 14), restriction to phylogenetic cluster analysis only (n = 18), or laboratory work (n = 11) and nosocomial infection by inadequately disinfected respiratory equipment (n = 1) made up only a minor reason for exclusion. In one study, Mtbc transmission between a 6-year-old child and his 46-year-old father, previously suggested to be a source case by using the VNTR method, could not be confirmed by more specific whole-genome sequencing (WGS) [13].

Finally, 39 studies published in peer-review journals were included for in-depth analysis and were deemed to be eligible to be included into the review (see references [14,15,16,17,18,19,20,21,22,23,24,25,26,27,28,29,30,31,32,33,34,35,36,37,38,39,40,41,42,43,44,45,46,47,48,49,50,51,52]). The studies are summarized in Appendix A.

### 3.1. Origin of Studies

The genotyping studies came from a total of 16 countries. Most were from the USA (13/39, 33.3%), the UK, Poland, Spain, and Germany (each 3/39, 7.7%). Further source countries were Denmark (2), Italy (2), South Africa (2), Latvia (1), Lebanon (1), Israel (1), Norway (1), Canada (1), Czech Republic (1), and Taiwan (1); see also the comprehensive Appendix A.

### 3.2. Study Participants and Study Design

A total of 2851 TB patients who were culture-confirmed and subsequently genotyped (any type) established the final body of evaluation of the 39 included studies (see Appendix A). Of those, 225 children aged below 15 years were included as cluster members. In Cronin’s study [23], the number of children among the 436 cluster members was not reported. In Marais’s study [41], the number of clusters among the 164 genotyped TB patients was not revealed, and Nordholm et al. [45] did not separate genotyping and epidemiological matches. The mean number of clustered children per study was 6.1, ranging from 1 to 44 children in each study.

Nineteen out of the total of 39 evaluated studies (49%) were outbreak analyses, and 6 studies (15%) were case reports. Twelve studies were cross-sectional, and two of these comprised retrospective data from nationwide registers [18,46]. Only two studies reported the data of a prospective community-based study [21,41]. Most studies (24/39 or 62%) utilized restriction fragment polymorphism (RFLP) and/or variable number tandem repeat (VNTR) (16/39 or 41%) for genotyping; in only three studies [15,32,33], whole genome sequencing (WGS) was used for all samples, with the latter applying the novel core genome multilocus sequence typing (cgMLST) scheme with a 12-allele cut-off.

### 3.3. Observational Period

The duration of case-finding of TB patients varied greatly: from 3 months [28] to 15 years [18] in which isolates were collected (see Appendix A). In 12 studies, however, transmission between TB cases were reported only for a maximum of 12 months. In 10 studies, there was no contact tracing performed or no information available on whether it was provided.

### 3.4. Drug Susceptibility

Seventeen studies reported susceptibility against first-line drugs. In nine studies, the cluster members had MDR-TB (multidrug-resistant tuberculosis); in nine studies, there was resistance against one of the first-line drugs, including Streptomycin. In three studies, two of them presenting the Beijing strain drug, resistance was not defined. In four studies, no information on drug susceptibility was provided.

### 3.5. Identification of Sources of Mtbc Transmission

In Cronin’s cross-sectional study on 1172 genotyped TB patients [23], all source cases and settings of transmission were identified for all instances of recent transmission except one: a 3-year-old. However, no information on the age of the source cases was provided. In Chin’s study [21], details on the known source cases of at least 12 children who became infected with Mtbc because the diagnosis of their sources had been delayed could not be found.

In Augustynowicz-Kopec’s study [17], which evaluated the Mtbc transmissions of 35 index cases to their household contacts, two children were included in two different clusters. In one of these clusters, the sputum-smear-positive mother was isolated two months later than her sputum-smear-negative 1-year-old daughter but it remains unclear whether the little child may have been infected by her mother or vice versa, as additional clinical and radiological information regarding the start of the mother’s TB disease were not available.

Only in three studies were children definitely identified as source cases by genotyping: In a one-point outbreak at a school in Leicester in 2001, a 13-year-old pupil transmitted Mtbc to nine other 7th-to 9th-year classmates [28]. In Curtis’s study [24], a 9-year-old boy infected his 36-year-old female guardian, and in Paranjothy’s study [47], a 9-year-old boy with pulmonary TB transmitted Mtbc to an 8-year-old boy whose culture of gastric lavage showed an identical MIRU-VNTR (Mycobacterial interspersed repetitive unit-variable number tandem repeat) pattern.

In all the remaining 34 studies, as far as a source case could be determined at all, the children’s adult contacts were either proven to be the source or judged as the most likely source of the infection.

## 4. Discussion

Contact investigations are a key component of the public health surveillance of TB in low-incidence countries: In 2020, in Germany, 49.0% of all pediatric TB cases were identified by contact tracing [53]. However, the recruitment of contacts in contact investigations is often incomplete. It is highly dependent on available public health management resources and the quality of personal questioning of contacts of reported index TB patients. Therefore, concomitant routine sampling and genotyping of Mtbc isolates in patients with culture-confirmed TB may help to confirm or disprove the suggested pathways of Mtbc transmission in both single person-to-person transmissions as well as in outbreaks [13,54]. Evaluating the pathways of Mtbc transmission to and coming from children by genotyping, however, is hampered by the paucibacillary nature of the TB disease in children. Consequently, Mtbc cultures may remain negative in children younger than 2 years, with probable pulmonary TB in up to 70% of cases, and in children aged between 5 and 15 years in up to 64% [5,55].

Our review of 39 genotyping studies of epidemiologic associations supports the current scientific opinion that children afflicted with pulmonary tuberculosis do indeed, if only rarely, transmit their disease to others: Only three children, all boys, have been genotypically proven to be at the origin of clusters: a 13-year-old as the origin of a school outbreak, a 9-year-old boy who transmitted Mtbc to his 36-year-old female guardian, and a 9-year-old boy who infected another 8-year-old boy. In one study, it remained unclear whether the sputum-smear-positive mother or the one-year-old sputum-smear-negative daughter was the source case.

The results of our review confirm that not every asymptomatic child with whom a diseased child had contact should be considered as a possible source of infection. On the other hand, the results also show that children do transmit TB although much less frequently than do adults and that this possibility must be considered in practice. This means that centrifugal contact tracing of children with pulmonary TB, especially in kindergartens and school classrooms, must—despite all assumptions of a low infectiousness—always be done carefully.

Our study, however, has some limitations that counsel caution before jumping to conclusions: Firstly, many molecular epidemiological studies found in our search were designed to assess risk factors for developing TB disease in children but did not focus on revealing epidemiological links between the single cluster members. Of note, 94 studies, nearly 2.5 as many as the 39 we were able to analyze, did not fulfill our inclusion criteria and had to be excluded from further analysis, thus leaving important information on transmission pathways behind. Thus, data exist in the literature on Mtbc transmission that we could not consider, as they are part of genotyping studies that were not designed to examine our question.

Secondly, in 12 of the 39 included studies (31%), the observational period was quite short, i.e., 12 months or less. Hence, a later development of TB cases in pediatric contacts may have taken place following closure of the studies available. Thirdly, regarding Mtbc transmissions in the community, it may be difficult to answer the question as to whether MTB (*Mycobacterium tuberculosis*) infections were part of an ongoing outbreak or restricted to the epidemiologically suggested close contact. This is even more so the case in short-term analyses utilizing older genotyping methods (RFLP, VNTR), which have limited specificity [56]. These limitations may be overcome by establishing nationwide prospective and long-term population-based genotyping studies using modern WGS procedures that are continuously reviewed in collaboration with the case-reporting public health authorities.

## 5. Conclusions

The results of our review suggest that children—as far as it has been verified by genotypically identical Mtbc isolates—usually do not transmit Mtbc to their contact persons. Due to probable underreporting of Mtbc transmissions as described above, however, these findings can only be considered preliminary. Further insights have to be achieved by investment in well-designed, prospective, population-based genotyping studies conducted at the national level over many years.

## Figures and Tables

**Figure 1 ijerph-20-01737-f001:**
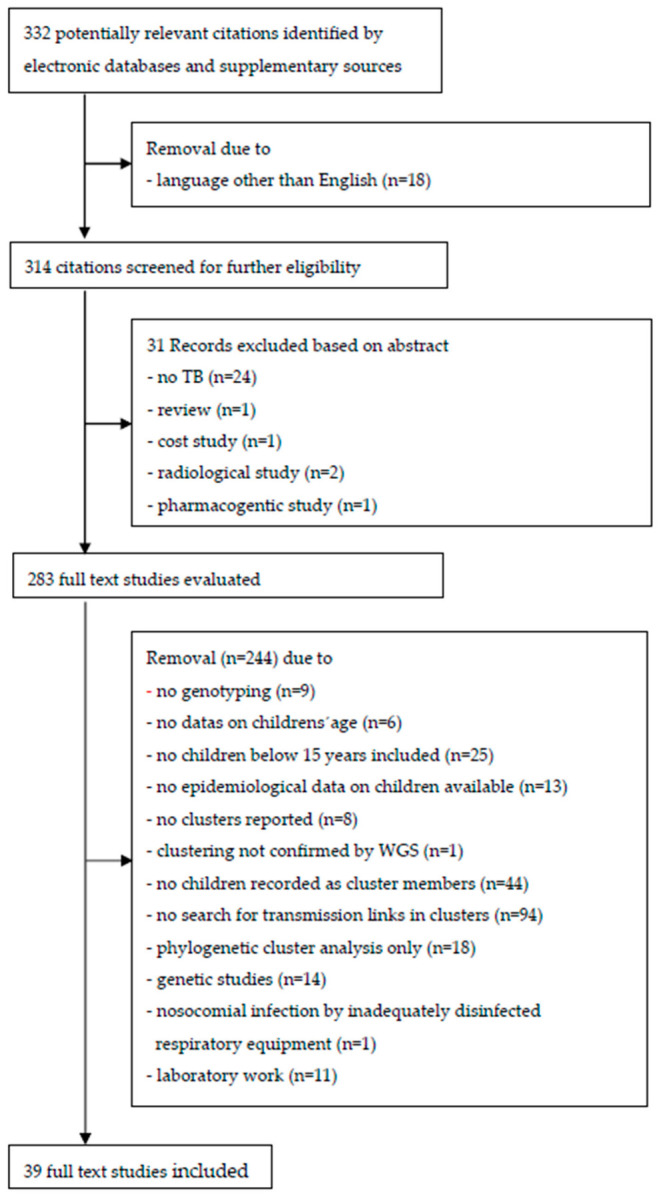
PRISMA flow diagram of study selection.

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
