# Peer review of "Pathways of TB Transmission in Children—A Systematic Review of Molecular Epidemiological Studies"

_ijerph, 2023, doi:10.3390/ijerph20031737_

Round 1
Reviewer 1 Report
Review IJERPH TB transmission in children. Diel and Nienhaus.
The goal of the study, to examine whether children under 15 years old transmit TB, is commendable, but there are several aspects that limit confidence in the findings. Transmission in the study is defined by clustering, which requires Whole Genome Sequencing that has only been applied progressively to TB molecular epidemiology studies over the past decade, and still today is not used in all epidemiological investigations. In addition, it is estimated that for those over 5 – 8 years, only 5% of individuals infected with TB will develop the disease within 2 years after infection, but the study only considers cases with TB disease who have positive cultures that were genome sequenced. In most areas of the world cultures are not routine, and even where all suspected TB patients are cultured and all isolates are genome sequenced, such as the UK, about 95% of those to whom the infection is transmitted will not be considered. Also, children under age 8 generally do not produce sputum and cultures, when mounted, are positive in only a minority of cases. It seems logical that these young children aren’t frequently transmitting TB. As a result, the study really concerns children between 8 and 15 years who have positive sputum cultures and whose isolates were genome sequenced. This would appear to limit the review to a very small universe of studies. In lines 59 – 62 the authors mention other studies documenting TB infection with tuberculin or IGRA tests but discard these because they cannot be genomicly proven to have spread from an index case. While this is a valid argument for perhaps the majority of studies, there are probably at least several studies describing TB infected classmates of a student with active TB, where the classmates don’t develop the disease but have a positive tuberculin or IGRA test. Although transmission from the index case can’t be genomicly proven, these school outbreak should be examined, and perhaps stratified by age, as it is likely that some of them could convincingly demonstrate transmission from children.
Minor points:
Line 58 Cavities is probably a better word than “caverns” .
Line 95 – What was the SNP cutoff used for defining clusters? How can one exclude that both cases were infected in the community as part of an ongoing community outbreak or both were infected from a family contact? Studies where the direction of transmission was not addressed were excluded from the analysis, further limiting its purview.
3.1 Origin
Most of the studies were from low TB incidence countries that can afford cultures and WGS, again limiting the scope and reliability of the study.
Line 153. How were 225 children members of 269 clusters? Some 44 children were members of 2 distinct clusters? This paragraph is not clear.
3.3 Observational period. Were the studies described here included in the analysis?
Lines 173-4 “reports having identified the sources of recent transmission except a 3-year old child but declines to report on their ages” The meaning is not clear.
Line 200 “Disprove” would be a better word than “falsify” .
Line 204. What ages are considered in the estimate that 70% of children have negative cultures?
Linees 208-9 “This implies that, in the absence of symptoms indicative of active disease, the stressful CXR procedure would be performed in addition to the LTBI testing that is currently done.” This sentence is hard to understand. CXR’s for children over 6 are not very stressful. LTBI involves some sort of needle under the skin (ppd or blood drawn), which is likely more stressful. In addition, these are diagnostic procedures for close TB contacts.
Lines 211 – 212. No one would argue that adults are not the main source of infection of children, but that is not the question under analysis, and (Lines 214 – 217) the 3 cases of transmission from children as young as 9, show that transmission is possible, but may not have been documented in WGS studies that were not designed to examine this question.
Author Response
The goal of the study, to examine whether children under 15 years old transmit TB, is commendable, but there are several aspects that limit confidence in the findings. Transmission in the study is defined by clustering, which requires Whole Genome Sequencing that has only been applied progressively to TB molecular epidemiology studies over the past decade, and still today is not used in all epidemiological investigations. In addition, it is estimated that for those over 5 – 8 years, only 5% of individuals infected with TB will develop the disease within 2 years after infection, but the study only considers cases with TB disease who have positive cultures that were genome sequenced. In most areas of the world cultures are not routine, and even where all suspected TB patients are cultured and all isolates are genome sequenced, such as the UK, about 95% of those to whom the infection is transmitted will not be considered. Also, children under age 8 generally do not produce sputum and cultures, when mounted, are positive in only a minority of cases. It seems logical that these young children aren’t frequently transmitting TB. As a result, the study really concerns children between 8 and 15 years who have positive sputum cultures and whose isolates were genome sequenced. This would appear to limit the review to a very small universe of studies. In lines 59 – 62 the authors mention other studies documenting TB infection with tuberculin or IGRA tests but discard these because they cannot be genomicly proven to have spread from an index case. While this is a valid argument for perhaps the majority of studies, there are probably at least several studies describing TB infected classmates of a student with active TB, where the classmates don’t develop the disease but have a positive tuberculin or IGRA test. Although transmission from the index case can’t be genomicly proven, these school outbreak should be examined, and perhaps stratified by age, as it is likely that some of them could convincingly demonstrate transmission from children.
Reply: Thank you for this detailed statement. However, we do not see any contradiction to the approach and results of our study; we would like to clarify a possible misunderstanding: The aim of the study was to clarify whether the prevailing doctrine, based predominantly on clinical- epidemiologic data, that children, especially children younger than 10 years, rarely transmit M. tb. Our intent was to objectify this doctrine, using molecular epidemiological fingerprint studies to provide evidence as to how epidemiological chains of infection run. In this respect, molecular fingerprint testing serves as a gold standard, beyond casuistic publications. Our work is the first systematic analysis to be carried out for this aspect of tuberculosis. That the results of our work would confirm the doctrine was not an a priori assumption for us. The results of our review suggest that not every asymptomatic child with whom a diseased child had contact should automatically be considered as a potential source of infection. On the other hand, the results of our review also show that children do transmit TB, although less frequently than do adults, and that this possibility must be considered in practice. This means that centrifugal environmental testing of sick children, especially in kindergartens and school classrooms, must be done carefully. We have now clarified this fact in the discussion.
Minor points:
Line 58 Cavities is probably a better word than “caverns” .
Reply: Thank you, replaced as requested
Line 95 – What was the SNP cutoff used for defining clusters? How can one exclude that both cases were infected in the community as part of an ongoing community outbreak or both were infected from a family contact? Studies where the direction of transmission was not addressed were excluded from the analysis, further limiting its purview.
Reply: Thank you for this objection: SNPs refer only to the WGS method used by 3 studies, of those the cut-off was not reported in two studies (reference 15 and 32). In the study by Kohl et al. (reference 33) a core genome multilocus sequence typing (cgMLST) scheme with a 12-allele cut-off was used. We have now added a sentence at the end of the section “3.2. Study participants and study design”. “Most studies (24/39, or 62%) utilized restriction fragment polymorphism (RFLP) and/or variable number tandem repeat (VNTR) (16/39, or 41%) for genotyping; in only 3 studies [15, 32, 33] whole genome sequencing (WGS) was used for all samples, the latter applying the novel core genome multilocus sequence typing (cgMLST) scheme with a 12-allele cut-off.”
Regarding MTB transmissions in the community, we agree and have now added the following sentence to the "limitations" comments providing a new reference:
“Thirdly, regarding Mtbc transmissions in the community: it may be difficult to answer the question as to whether Mtbc infections were part of an ongoing outbreak or restricted to the epidemiologically suggested close contact. This is even more so the case in short-term analyses utilizing older genotyping methods (RFLP, VNTR) with limited specificity [57].”
3.1 Origin
Most of the studies were from low TB incidence countries that can afford cultures and WGS, again limiting the scope and reliability of the study.
Reply: You are right that more studies come from low incidence countries. However, we see no limitation to the applicability of our findings for high-incidence environments, the characteristic infectiousness of TB being the same in high-incidence countries as it is in low-incidence areas.
Line 153. How were 225 children members of 269 clusters? Some 44 children were members of 2 distinct clusters? This paragraph is not clear.
Reply: Thank you for this note: The number of 269 clusters refers to the total number of all clusters from the respective studies, and not all clusters had children as patients. We have changed the misleading wording and deleted the "269".
Sin's study included 44 children in the clusters listed there, representing the upper bound of the range of children included in all cluster studies We have now rephrased the sentence: “The mean number of clustered children per study was 6.1, ranging from 1 to 44 children in each study.
3.3 Observational period. Were the studies described here included in the analysis?
Reply: Yes, all studies were included. We have updated our text to now refer to the comprehensive Table in brackets in the text.
Lines 173-4 “reports having identified the sources of recent transmission except a 3-year old child but declines to report on their ages” The meaning is not clear.
Reply: Yes, we agree and have now rephrased the sentence: “In Cronin's cross-sectional study on 1,172 genotyped TB patients [23] all source cases and settings of transmission were identified for all instances of recent transmission except one, a 3-year-old. However, no information on the age of the source cases was provided.”
Line 200 “Disprove” would be a better word than “falsify” .
Reply: Thank you, replaced as suggested.
Line 204. What ages are considered in the estimate that 70% of children have negative cultures?
Reply: Thank you, we have again reviewed the current literature and replaced reference 5 (which only cites reference 56) by the study of DiNardo et al. (DiNardo AR, Detjen A, Ustero P, Ngo K, Bacha J, Mandalakas AM. Culture is an imperfect and heterogeneous reference standard in pediatric tuberculosis. Tuberculosis (Edinb). 2016 Dec;101S:S105-S108).
The sentence now reads: Consequently, Mtbc cultures may remain negative in children younger than 2 years with probable pulmonary TB in up to 70% of cases, and in children aged between 5 and 15 years in up to 64% [5,56].
Lines 208-9 “This implies that, in the absence of symptoms indicative of active disease, the stressful CXR procedure would be performed in addition to the LTBI testing that is currently done.” This sentence is hard to understand. CXR’s for children over 6 are not very stressful. LTBI involves some sort of needle under the skin (ppd or blood drawn), which is likely more stressful. In addition, these are diagnostic procedures for close TB contacts.
Reply: According to our personal experience, an X-ray examination, at least in small children, which usually has to be performed in pediatric clinics, is stressful for children and their parents. However, this is indeed also true for the other examinations (Mantoux, IGRA), especially if they are performed without a clear indication. We have therefore deleted the sentence (and the preceding sentence), as it is not pertinent to the discussion of the question.
Lines 211 – 212. No one would argue that adults are not the main source of infection of children, but that is not the question under analysis, and (Lines 214 – 217) the 3 cases of transmission from children as young as 9, show that transmission is possible, but may not have been documented in WGS studies that were not designed to examine this question.
Reply: Thank you for this comment. Indeed, the question under analysis is whether children with pulmonary TB may frequently infect other persons, adults as well as other children. Therefore we have rephrased the sentence: “Our review of 39 genotyping studies of epidemiologic associations supports the current scientific opinion that children afflicted with pulmonary tuberculosis do indeed, if only rarely, transmit their disease to others.”
Reviewer 2 Report
I read this paper with great interest and joy.
For epidemiologists working on tuberculosis control, this is a question of particular interest. Do children transmit tuberculosis? Older, younger? Do we have to trace their contacts?
Little is known about children as sources of tuberculosis transmission. This interesting research took a lot of effort to consider this particular question. I believe that this issue of tuberculosis control is very relevant and has policy implications.Methodology of this study is appropriate. The procedure for selecting studies is described in detail, the results are clearly presented, using a qualitative approach. However, of the 39 studies finally selected, only three studies definitively identified children as source cases. Not such a surprise for a 13-year-old student, but two source cases among 9-year-old children deserve additional attention. However, this is not enough to draw an appropriate conclusion. Chapter 1. Introduction, the abbreviation used in line 61- IGRA, needs to be explained, although it is well known. In addition, the manuscript would benefit from editing by a native speaker.
Author Response
I read this paper with great interest and joy.
For epidemiologists working on tuberculosis control, this is a question of particular interest. Do children transmit tuberculosis? Older, younger? Do we have to trace their contacts?
Little is known about children as sources of tuberculosis transmission. This interesting research took a lot of effort to consider this particular question. I believe that this issue of tuberculosis control is very relevant and has policy implications.Methodology of this study is appropriate. The procedure for selecting studies is described in detail, the results are clearly presented, using a qualitative approach.
- However, of the 39 studies finally selected, only three studies definitively identified children as source cases. Not such a surprise for a 13-year-old student, but two source cases among 9-year-old children deserve additional attention. However, this is not enough to draw an appropriate conclusion.
Reply: Thank you, we fully agree and have addressed this statement by adding a sentence in the Discussion section.
- Chapter 1. Introduction, the abbreviation used in line 61- IGRA, needs to be explained, although it is well known.
Reply: Thank you, done as suggested.
- In addition, the manuscript would benefit from editing by a native speaker.
Reply: The manuscript has anew been seen by a native speaker.
Reviewer 3 Report
In the present study, the authors have studied the epidemiology of the pathways of TB transmission in children.
1. Rewrite the introduction as it doesn't justify the title of the manuscript.
2. Illustrate the findings to make the article more interesting and to reach more readers.
3. In line 222- 224, authors have mentioned that due to there inclusion criterias limitation they have to leave 94 reports that have valuable information. I would suggest to either redefine the inclusion criteria or illustrate the major findings in the excluded articles.
4. Considering lower number of case that studies children age below 10, it's hard to justify the conclusion. To make it more convincing provide enough data.
Author Response
Reviewer 3
In the present study, the authors have studied the epidemiology of the pathways of TB transmission in children.
- Rewrite the introduction as it doesn't justify the title of the manuscript.
Reply: Thank you, we have now shortened the introduction and improved it in line with the comments of the other reviewers.
- Illustrate the findings to make the article more interesting and to reach more readers.
Reply: Thank you for this proposal but, as this is a systematic review we have focused on presenting the quality of the studies and the study results as clearly as possible in the table that serves as an appendix here.
- In line 222- 224, authors have mentioned that due to there inclusion criterias limitation they have to leave 94 reports that have valuable information. I would suggest to either redefine the inclusion criteria or illustrate the major findings in the excluded articles.
Reply: Thank you, but the 94 reports that we have had to exclude did not contribute valuable information with respect to the aim of the study because their study designs precluded a focus on children. Therefore, it is not possible to present the results that are relevant for our question but unfortunately missing. However, we have once again explicitly emphasized in the discussion under “limitations” that due to the inadequacies of these studies, valuable information may have been lost and transmissions may have been overlooked.
- Considering lower number of case that studies children age below 10, it's hard to justify the conclusion. To make it more convincing provide enough data.
Reply: As our systematic review includes a total of 2,851 TB patients who were culture-confirmed and subsequently genotyped we do not feel that the data basis is principally insufficient. We want to note politely that we did not hypothesize any new, independent conclusions that would contradict the previous scientific knowledge; the results of our study simply confirm the current scientific doctrine that children aged below 10 years have a low degree of infectiousness. Despite this, we follow your view as we had already stated in our conclusions that due to the low number of studies our results – as far as the genotypic perspective is taken – can only be considered preliminary and we strongly call for more long-term population-based studies on the topic.